# Role of Biomarkers in FLT3 AML

**DOI:** 10.3390/cancers14051164

**Published:** 2022-02-24

**Authors:** Jiao Wei, Ai-Min Hui

**Affiliations:** 1Fosun Pharma USA Inc., Boston, MA 02421, USA; fnunitika@fosunpharma.com (N.); weijiao@fosunpharma.com (J.W.); 2Shanghai Fosun Pharmaceutical Industrial Development, Co., Ltd., Shanghai 200233, China

**Keywords:** acute myeloid leukemia, FLT3 inhibitors, FLT3-ITD, FLT3-TKD, biomarkers, resistance

## Abstract

**Simple Summary:**

Genetically heterogeneous disorder acute myeloid leukemia (AML) is marked by recurring mutations in FLT3. Current FLT3 inhibitors and other emerging inhibitors have helped in the improvement of the quality of standard of care therapies; however, the overall survival of the patients remains static. This is due to numerous mutations in FLT3, which causes resistance against these FLT3 inhibitors. For effective treatment of AML patients, alternative approaches are required to overcome this resistance. Here, we will summarize the biomarkers for FLT3 inhibitors in AML, as well as the alternative measures to overcome resistance to the current therapies.

**Abstract:**

Acute myeloid leukemia is a disease characterized by uncontrolled proliferation of clonal myeloid blast cells that are incapable of maturation to leukocytes. AML is the most common leukemia in adults and remains a highly fatal disease with a five-year survival rate of 24%. More than 50% of AML patients have mutations in the FLT3 gene, rendering FLT3 an attractive target for small-molecule inhibition. Currently, there are several FLT3 inhibitors in the clinic, and others remain in clinical trials. However, these inhibitors face challenges due to lack of efficacy against several FLT3 mutants. Therefore, the identification of biomarkers is vital to stratify AML patients and target AML patient population with a particular FLT3 mutation. Additionally, there is an unmet need to identify alternative approaches to combat the resistance to FLT3 inhibitors. Here, we summarize the current knowledge on the utilization of diagnostic, prognostic, predictive, and pharmacodynamic biomarkers for FLT3-mutated AML. The resistance mechanisms to various FLT3 inhibitors and alternative approaches to combat this resistance are also discussed and presented.

## 1. Introduction

Acute myeloid leukemia (AML) is a hematological malignancy that accounts for the most common leukemia that occurs in adults. In the US, there were 20,240 cases and 11,400 deaths due to AML in 2021. The five-year relative survival rate for AML patients is 29.5% [1]. The incidence of AML increases with age, whereby there are 1.3 cases per 100,000 of the population who are under 65 years old, whereas there are 12.2 cases per 100,000 of the population who are 65 years old and above [2]. Several treatment options have improved the survival of younger patients, but the mortality remains high for elderly patients [2,3].

AML is characterized by clonal proliferation of poorly differentiated cells of hematologic origin. These cells are genetically altered with recurrent deletions, amplifications, point mutations, and rearrangements [4,5].

The human flt3 (FMS-like tyrosine kinase 3) gene is located on chromosome 13q12 and has 24 exons [6]. It encodes a membrane-bound glycosylated protein with a molecular weight of 160 kDa, along with a non-glycosylated isoform which is 143 kDa and not associated with the plasma membrane [7]. FLT3 is a transmembrane protein that encodes for proto-oncogene FLT3. It is a member of the class III receptor tyrosine kinase family and plays an important role in the regulation of the hematopoiesis [8]. The structure of FLT3 consists of four regions: (i) an N-terminal, extracellular region consisting of five immunoglobulin domains involved in ligand binding; the proximal domain is involved in receptor dimerization, (ii) a transmembrane domain, (iii) a juxta membrane domain (JM), and (iv) an intracellular, C-terminal region with a split-kinase domain. The two substructures of this domain are called N-lobe and C-lobe, which are connected by an inter-kinase domain. These lobes consist of a TKD and are also indicated as the first tyrosine kinase (TK1) and second tyrosine kinase (TK2) domain, respectively [9] (Figure 1).

The extracellular region contains a binding domain with a high affinity for its ligand (FLT3 ligand or FL). FL is expressed by most tissues, including the spleen, thymus, and bone marrow; however, the highest expression is seen in peripheral blood mononuclear cells [7].

Once the FL binds to FLT3, it induces receptor dimerization and conformational changes. Subsequently, FLT3 autophosphorylation activates intracellular signaling cascades that control cell proliferation, differentiation, and survival [7,9,10]. The kinase activity of the FLT3 receptor is negatively modulated by tyrosine phosphatase that dephosphorylates the JM domain. Thus, the frequency of FLT3 production, its degradation, and the downstream effects are regulated by a complex feedback loop for the normal activity of the receptor [7].

The most common mutations in FLT3 include FLT3 internal tandem duplication (FLT3-ITD^mut^), which is detected in 25% of patients, and point mutations in the tyrosine kinase domain (FLT3-TKD^mut^) that are detected in 7–10% of the patients [11]. These mutations result in the overexpression or constitutive activation of the tyrosine kinase receptor and the downstream proliferative signaling pathways. In addition, FLT3-ITD^mut^ potently activates STAT5, which activates cyclin D1, c-myc, and Pim-2; the activation of these proteins results in the accelerated growth of leukemic cells [7,12]. FLT3-TKD^mut^ consists mainly of missense point mutations, deletions, or insertions in the tyrosine kinase domain of FLT3. The most frequent point mutations are primarily seen in the activation loop in amino acid residues D835, I836, Y842, and some in the TKD1, including the residues N676 and F691 [13].

## 2. Biomarkers for FLT3 Inhibition in AML

A biomarker is a characteristic that is a measurable indicator of a biological process or response to an intervention. Molecular biomarkers are valuable in providing information about the biological behavior of the AML. These biomarkers can be classified into various categories, including diagnostic, predictive, prognostic, and pharmacodynamic biomarkers, based on their putative applications [14].

Diagnostic biomarkers are used to confirm the presence of a disease and aid in the identification of individuals with a disease subtype. These biomarkers are used to identify people with a disease [14]. For example, in the case of AML, gene rearrangements, gene fusions, and chromosomal translocations are used in the diagnosis [15].

Predictive biomarkers are used to identify the likelihood of response or lack of response to a particular therapy. These biomarkers help in the identification of patients most likely to benefit from a given treatment and spare other patients from the toxicities of ineffective therapies [14]. NPM1 mutations and the FLT3-ITD allelic ratio (AR) are candidate predictive biomarkers in FLT3 AML [16].

Prognostic biomarkers are used to identify the likelihood of a clinical event, disease recurrence, or progression in patients who have the disease [14]. Mutations in the FLT3 gene, such as FLT3-ITD, confer a poor prognosis in AML patients [11]. Pharmacodynamic biomarkers depict the biological response to a medical product or environmental agent in an individual. Such biomarkers are useful for clinical practice and therapeutic development [14]. Various molecular markers, such as phosphorylation, and immune markers have been used in various studies [17,18].

All these biomarkers are important because of their high clinical importance, and their expression can reveal the disease evolution in real time [19]. So far, several biomarkers have been identified by various studies and clinical trials of FLT3 inhibitors, which are discussed in the sections below.

## 3. Diagnostic Biomarkers

Various diagnostics have been developed for detecting AML, including morphological, immunophenotyping, and gene fusion screening [20]. For morphological diagnostics, bone marrow smears are examined for myeloblasts, monoblasts, and megakaryoblasts in the blast cells using Wright-Giemsa stains. Immunophenotyping using flow cytometry is used to determine the lineage of leukemia cells [21]. In AML patients, leukemic cells express early, hematopoiesis-associated antigens (CD34, CD38, CD117, HLA-DR) and lack markers of myeloid and monocytic maturation (NSE, CD11c, CD14, CD64) [15,22,23,24]. Similarly, cytogenetic abnormality can be detected in 50% to 60% of newly diagnosed AML patients. The majority of AML patients have nonrandom chromosomal translocations that often lead to gene rearrangements. [25,26,27]. The World Health Organization (WHO) recognizes recurrent translocations and inversions in AML [28,29] (Table 1). Gene rearrangements, gene fusions, and loss of chromosomes are detected using fluorescence in situ hybridization (FISH) and reverse transcriptase–polymerase chain reaction (RT-PCR) [30,31]. These include gene fusions in RUNX1-RUNX1T1 (runt-related transcription factor 1), CBFB-MYH11 (core-binding factor subunit beta–myosin heavy chain 11), acute promyelocytic leukemia (APL) with PML-RARA (promyelocytic leukemia/retinoic acid receptor alpha), MLLT3-KMT2A (mixed-lineage leukemia translocated to chromosome 3- lysine methyltransferase 2A), DEK-NUP214 (DEK oncogene–nucleoporin 214), and an inversion that repositions a distal GATA2 enhancer to activate MECOM expression. BCR-ABL1 is added to recognize that these cases may benefit from tyrosine kinase inhibitor therapy [28,29,32,33]. Finally, for AML diagnosis, testing for mutations in three genes—FLT3, NPM1 (nucleophosmin 1), and CEBPA (CCAAT/enhancer binding protein (C/EBP) alpha)—is recommended [34,35,36]. Additional genes with varying gene mutation frequency in AML patients include mixed-lineage leukemia (MLL), neuroblastoma RAS (NRAS), Wilms’ tumor type 1 (WT1), v-KIT, runt-related transcription factor (RUNX1), and iso-citrate dehydrogenase (IDH1) [37,38,39,40,41,42].

Interestingly, recent studies indicated that circulating micro RNAs (miRNAs) can be utilized as diagnostic biomarkers for AML. A study identified six serum miRNAs (miR-10a-5p, miR-93-5p, miR-129-5p, miR-155-5p, miR-181b-5p, and miR-320d) which were specifically upregulated in the serum of AML patients using a next-generation sequencing approach [43,44].

Four miRNAs (let-7b, miR-128a, miR-128b, and miR-223) were used for the diagnosis of AML with 97% accuracy and analyzed using RT-PCR. miR-142-3p and miR-29a can also be used as diagnostic biomarkers for AML [45]. Interestingly, miR-424 was downregulated in AML patients with NPM1 mutation regardless of FLT3 mutation, whereas miR-155 was upregulated in patients with FLT3-ITD regardless of the NPM1 mutation [46].

These studies suggest that miRNAs from serum or blood samples can be effective diagnostic biomarkers for AML patients.

## 4. Predictive Biomarkers

There have been multiple FLT3 inhibitors in clinical trials, but predictive biomarkers remain undiscovered. In a recent study aimed at identifying gene expression changes associated with FLT3 mutation in AML patients, the transcriptomic patterns of six different cohorts of AML patients were analyzed, and a FLT3-mutation-like pattern was highly enriched in NPM1 and DNMT3A mutants. In addition, FLT3-like patterns consisted of numerous homeobox (HOX) genes [47].

Based on the FLT3 mutations, companion diagnostics were generated that tested for a predictive biomarker [48]. These tests classified patients into responders and non-responders and directly equated them to the administration of a drug [49].

One such FDA-approved companion diagnostic test was the LeukoStrat CDx FLT3 mutation assay. This is a PCR-based, in vitro diagnostic test that detects ITD and TKD mutations D835 and I836 from the genomic DNA extracted from the mononuclear cells from peripheral blood or bone marrow aspirates of AML-diagnosed patients [50].

This test was used with FLT3 inhibitors, including midostaurin, gilteritinib, and quizartinib [51].

Similarly, co-occurrence of mutations in FLT3 with national comprehensive cancer network (NCCN)-listed gene mutations were used as predictive biomarkers [52,53]. Co-occurrence of mutations in monoallelic, CCAAT/enhancer-binding protein alpha (moCEBPA) with FLT3-ITD/TKD led to a poor prognosis. Mutations in NPM1, DNMT3A, and FLT3-ITD were identified at higher rates in patients with intermediate-risk cytogenetics [54,55]. It was seen that a group of AML patients with FLT3 plus NPM1 and/or DNMT3A mutations shared a similar transcriptomic background [47]. The revised 2017 WHO classification has myeloid neoplasms with germline mutations in RUNX1, CEBPA, DDX41 (DEAD-box helicase 41), RUNX1, GATA2 (GATA binding protein 2), ETV6 (ETS variant transcription factor 6), SRP72 (signal recognition particle 72), and ANKRD26 (ankyrin repeat domain 26) as markers of AML predisposition [29,56,57].

Another study identified that the response to gilteritinib and crenolanib among relapsed FLT3^mut^ AML patients is higher in patients with mutations in NPM1 or DNMT3A and particularly in those with both genes mutated [58,59]. When FLT3-ITD leukemias with mutations in NPM1 or DNMT3A are treated with quizartinib, the cell differentiation effect predominates over the cytotoxic mechanism [60]. Additionally, a long non-coding RNA (lnc RNA) expression profile using RNA-seq identified that lncRNA RP11-342 M1.7, lncRNA CES1P1, and lncRNA AC008753.6 serve as predictive biomarkers for AML risk [61].

## 5. Prognostic Biomarkers

FLT3 is widely overexpressed and the most frequently mutated gene in both pediatric and adult patients with AML [62]. Higher expression of FLT3 results in poor overall survival (OS) in AML patients, as seen in the cancer genome atlas (TCGA) dataset analyzed by GEPIA. The hazard ratio is 1.8 for high-FLT3-expressing patients, indicating that these patients have a ~2 times greater chance of dying compared to the low-FLT3-expressing AML patients [63] (Figure 2).

FLT3 ligand (FL) is detectable during homeostasis and is increased in hypoplasia. FL is markedly elevated upon the depletion of the hematopoietic stem or progenitor cells. However, in FLT3^+^ AML, the levels of FL fall to undetectable levels. It was observed that, after the induction of chemotherapy, FL levels are restored in patients with complete remission but not in patients with refractory disease. FL levels were measured in a randomized study with lestaurtinib where it was seen that patients achieving complete remission (CR) had higher FL levels after the completion of the therapy followed by a normal range after recovery. However, patients with refractory disease had a transient increase in FL levels followed by rapid depletion [64]. Thus, FL levels have the potential to emerge as prognostic biomarkers to guide clinical decisions.

The presence or absence of specific gene mutations can be utilized to classify AML patients and determine their prognosis. The NCCN AML prognostic stratification system listed FLT3, NPM1, CEBPA, IDH1/2, DNMT3A (DNA methyltransferase 3A), KIT, TP53 (tumor suppressor 53), RUNX1, and ASXL1 (ASXL transcription factor) gene mutations for the classification of the AML patient population [65,66]. Mutations of NRAS and IDH2 occur in FLT3-independent clones, but TET2 and IDH1 co-occur in FLT3-mutant clones [67].

Mutations in the FLT3 gene are of prognostic value for detecting AML in patients. The most common FLT3 mutations (FLT3^mut^) occur in the JM domain internal tandem duplications, FLT3-ITD^mut^, or in the tyrosine kinase domain, FLT3-TKD^mut^. FLT3-ITD^mut,^ are in-frame mutations consisting of duplications of 3–400 base pairs which lead to an elongated JM. This results in constitutive activation of the FLT3 receptor and the downstream signaling (Figure 3) [68,69].

The prognostic value of FLT3-ITD is determined by various factors, including the allele ratio (AR), ITD length, karyotype, insertion site, and co-mutations (NPM1) [11,70]. AR is the ratio of ITD-mutated alleles to wild-type alleles (FLT3-ITD/FLT3 wild-type). Similarly, variant allele frequency is determined by the ratio of ITD-mutated alleles to ITD-mutated and wild-type alleles. The European Leukemia Net (ELN) identified a value of 0.5 as a cut-off to distinguish between low and high AR [71]. FLT3-ITD insertion (AR > 0.51) is associated with an unfavorable, relapse-free survival, RFS (*p* = 0.0008) and OS (*p* = 0.004) [72]. However, a recent study depicted that the size of FLT3-ITD mutations has no prognostic impact on the overall survival, relapse, or complete remission rate among newly diagnosed AML patients treated with chemotherapy [73].

Favorable relapse risk and OS was seen with the occurrence of co-mutations NPM1, along with FLT3^mut^, in young adult AML patients [74]. In patients with concurrent NPM1^mut^, the OS and relapse risk were comparable between FLT3 wild-type and FLT3-ITD^mut^ (AR < 0.5), but worse when AR ≥ 0.5 [75]. Among patients with NPM1 wild-type, all FLT3-ITD^mut^ patients had an increased risk of relapse and inferior OS, regardless of the AR. The European Leukemia Net (ELN) guidelines categorize FLT3-ITD^mut^ AML into three categories: favorable (NPM1^mut^ with FLT3 wild-type or NPM1^mut^ with FLT3-ITD AR < 0.5), intermediate (NPM1^mut^ with FLT3-ITD AR > 0.5 or NPM1WT with FLT3-ITD AR < 0.5), and adverse (NPM1WT with FLT3-ITD AR > 0.5) [76]. Although the AR ratio is predictive of the severity of AML in the patients, a strict threshold cannot be established for clinical decision making. This is because the current assays are not optimized, and there is a high intrasample variability [77].

FLT3-TKD mutations have prognostic value in the overall AML patient population, but the impact of FLT3-TKD^mut^ AR remains obscure. However FLT3-TKD^mut^ has a high incidence in co-occurrence with mutations in NPM1, CEBPA, and NRAS [78]. These TKD mutations can be identified and detected using next-generation sequencing (NGS). Additionally, computational, biology-based algorithms, such as Pindel, show high sensitivity and specificity in detecting these gene alterations [79].

## 6. FLT3 Inhibitors in Clinical Trials and Development

Since FLT3 mutations lead to dysregulation of cell proliferation pathways, inhibiting FLT3 signaling using small molecule inhibitors is a viable strategy for AML patients [80].

FLT3 inhibitors can be classified into type I and type II based on their mechanism of interaction with FLT3 (Table 2). Type I inhibitors bind to the gatekeeper domain near the activation loop or the ATP binding site on the receptor regardless of its conformation; however, type II inhibitors bind to the hydrophobic region adjacent to the ATP binding site on the receptor in its inactive conformation. As a result, type I inhibitors can inhibit FLT3 with both ITD and TKD, but type II inhibitors can only inhibit FLT3 with ITD and not TKD [51]. Type I inhibitors include FN-1501, sunitinib, lestaurtinib, midostaurin, crenolanib, and gilteritinib, while type II inhibitors include sorafenib, quizartinib, ponatinib, and pexidartinib [81,82,83,84].

FLT3 inhibitors can also be classified into first generation and second generation based on their specificity for FLT3 (Table 2). First-generation inhibitors lack specificity for FLT3. They can bind to multiple receptor tyrosine kinases (RTKs) and inhibit several targets downstream of the FLT3 signaling pathway and parallel pathways, thus, providing a broad range of efficacy in AML patients. Second-generation inhibitors are more specific and only target FLT3. As a result, they are expected to have fewer off-target effects and toxicities. First-generation inhibitors include sunitinib, sorafenib, midostaurin, and lestaurtinib. Some second-generation inhibitors include quizartinib, crenolanib, and gilteritinib [11,96,97].

Currently, there are only three FDA-approved FLT3 inhibitors, sorafenib, midostaurin, and gilteritinib, for use in the U.S. Of these, only two are approved for AML indication: midostaurin, along with chemotherapy, and gilteritinib [98]. Midostaurin is a multi-targeted kinase inhibitor with activity against both FLT3-ITD and FLT3-TKD, along with induction chemotherapy with cytarabine and daunorubicin; however, it has limited clinical efficacy as a single agent [87,99]. On the other hand, gilteritinib was recently approved by the FDA with activity against FLT3-ITD, FLT3-TKD, and FLT3, non-canonical mutations in relapsed and refractory (R/R), FLT3-mutated AML patients as a monotherapy [100]. Sorafenib is not approved for the treatment of AML, but off-label use in 13 patients showed improved clinical outcomes in FLT3-ITD^mut^ AML patients [101].

Several other FLT3 inhibitors are in the early and late stages of clinical development. Quizartinib is the most potent and selective type II inhibitor and crenolanib is a potent type I inhibitor in the late stages of clinical development (Table 2).

## 7. Indicators for the Efficacy of FLT3 Inhibitors (Pharmacodynamic Biomarkers)

Biomarkers that predict the efficacy of FLT3 inhibitors have important applications in clinical care. Activation of FLT3 triggers the phosphorylation and activation of downstream signal transduction pathways including PI3K/AKT/mTOR, RAS/RAF/MAPK and JAK/STAT (Figure 3). Autophosphorylation of the FLT3 receptor has proven to be an excellent biomarker for its activation, and loss of this autophosphorylation is an indication of successful inhibition. The degree of phosphorylation can be quantified directly in the circulating blast cells or the plasma from the patients. Phospho-FLT3 levels can be measured using an enzyme-linked immunosorbent assay (ELISA) and plasma inhibitory activity (PIA) assay. Decreased phosphorylation of FLT3 is associated with clinical activity in patients administered with gilteritinib, midostaurin, and lestaurtinib [18,89,102]. Recently, SEL24/MEN1703, a dual PIM/FLT3 kinase inhibitor, underwent clinical trials for AML patients. This study tested phospho-inhibition of S6, 4-EBP1, and STAT5 as their phosphorylation levels are controlled by both PIM1/2 and FLT3. Preclinical studies identified that S6 phosphorylation (pS6) was at its maximum 4 h post drug treatment; hence, pS6 was chosen as a biomarker for this dual kinase inhibitor (Table 3). pS6 was measured from the whole blood and bone marrow of the patients administered with the drug-using flow cytometry [103,104]. Another preclinical study evaluated the potential use of follistatin (FST) as a pharmacodynamic biomarker. It was seen that, in FLT3-ITD patients treated with quizartinib, serum FST levels significantly decreased but resurged during relapse [105].

Another study identified that the expression of immune checkpoint markers CD155 and CD112 (using flow cytometry and real-time PCR) was specifically downregulated upon treatment with gilteritinib and quizartinib in FLT3-mutated cell models. Thus, CD155 and CD112 have the potential to serve as PD biomarkers for FLT3-ITD AML patients [17].

## 8. Resistance to FLT3 Inhibitors

Although FLT3 inhibitors show response in AML patients, the duration of this response is short-lived due to primary and acquired resistance. The most common mechanism of acquired resistance in patients is due to on-target mutations in the tyrosine kinase domain. F691L and D835 are frequently occurring FLT3 gatekeeper mutations. These mutations hinder the drug binding which results in an active kinase conformation unfavorable to interaction with FLT3 inhibitors [100,106]. This resistance mechanism was reported for type II inhibitors, including quizartinib and sorafenib. Both gilteritinib and crenolanib had preclinical and clinical activity against FLT3 D835 mutations, but they had limited activity against the F691L mutations [83,107,108,109]. However, pexidartinib and ponatinib had activity against F691L mutations in preclinical models [98]. Recently, another FLT3 inhibitor, FF-10101, displayed significant activity against F691L and D835 both in vitro and in vivo [110].

In addition, FLT3-ITD mutations contribute to resistance to the FLT3 inhibitors. This is because FLT3-WT is sensitive to FLT3 ligand and resistant to FLT3 inhibitors. FLT3-ITD has a WT sequence; therefore, it contributes to the resistance [111,112,113].

It was also observed that high levels of FL in the bone marrow microenvironment during induction and consolidation therapy can lead to activation of the FLT3-MAPK pathway and provide a survival signal to the blast cells even in the presence of FLT3 inhibitors [111]. Some preclinical studies also demonstrated that CYP3A4 in the bone marrow stromal cells also leads to FLT3-TKI resistance [114].

Acquired resistance due to non-overlapping, secondary mutations is caused by different FLT3 inhibitors. In-vitro-based studies demonstrated that SU5614 produced TK2 changes in D835 exclusively; however, midostaurin produced mutations in TK1 at N676. In addition, sorafenib produced resistant mutations in TK1 (F691L) and TK2 (Y842). These mutations led to different drug responses. While TK2 mutations were sensitive to midostaurin, sunitinib, and sorafenib, TK1 mutations had a differential response to SU5614, sorafenib, and sunitinib but impaired response to midostaurin [115].

Additional resistance mechanisms were seen where FLT3-TKI resistant cells became FLT3-independent due to the activation of parallel signaling pathways, including Ras/MEK/MAPK and PI3K/Akt, which compensate for cell survival signals when FLT3 is inhibited. Additionally, activating mutations in the Ras/MAPK pathway, including NRAS, PTPN11, KRAS, and CBL, were of common occurrence in gilteritinib and crenolanib resistance [116,117,118].

## 9. Fighting FLT3 Inhibitor Resistance and Future Approaches

Given that FLT3 inhibitors present limited efficacy due to the reasons mentioned above, alternate approaches are required to cure FLT3-AML patients. One of the approaches is the use of combination therapies of FLT3 inhibitors with other agents to enhance their efficacy and identify synergistic drug combinations. A combination of sorafenib with vorinostat (histone deacetylase inhibitor) was seen to be effective against FLT3 AML in an early-phase clinical trial [106]. Additionally, triple combinations of sorafenib, vorinostat, and bortezomib (proteasome inhibitor) were effective in early-phase clinical trials [119]. Similarly, a combination of sorafenib and azacytidine (DNA methyltransferase inhibitor) was effective for patients with FLT3-ITD and relapsed AML [120]. Combination therapies with signaling proteins downstream of the FLT3 pathway are another viable approach to overcome FLT3-inhibitor resistance. It was seen that pimozide (STAT5 inhibitor) is synergistic with midostaurin and sunitinib in FLT3-ITD patients in early-phase clinical trials [121]. Recently developed PIM1/FLT3 dual inhibitor SEL23/MEN1703, which targets Pim-1 (a kinase downstream of FLT3) and FLT3 together, is currently undergoing clinical trials [103]. Another study is assessing the safety of everolimus (mTOR inhibitor) with midostaurin in patients with relapsed and refractory AML (NCT00819546). A promising approach is targeting heat shock proteins, including Hsp40, Hsp70, and Hsp90 [122,123,124,125]. Hsp70 inhibition suppresses the proliferation of FLT3-ITD-positive and drug-resistant AML cells via the induction of proteasome-mediated degradation of FLT3-ITD [126]. Co-treatment of midostaurin with 17-AAG (Hsp90 inhibitor) attenuated phospho-FLT3 and induced apoptosis in human acute leukemia cells MV4-11 [127].

Some studies used anti-FLT3 monoclonal antibody (mAb) treatment for FLT3-TKI-resistant clones, but it was found to be a cytotoxic [128,129,130]. An alternative approach would be using FLT3 inhibitors with anti-FLT3 antibodies and/or inhibitors of the downstream signaling proteins of FLT3 [128].

Additionally, chimeric antigen receptor (CAR) T cells are an emerging novel therapeutic approach to target FLT3 AML. CD8+ and CD4+ T-cells expressing an FLT3-specific chimeric antigen receptor (CAR) were efficacious in vitro. Treatment with crenolanib enhanced the surface expression of FLT3 on FLT3-ITD AML cells which led to increased recognition by FLT3-CAR T cells in preclinical studies [131].

## 10. Conclusions

FLT3 inhibitors show promising efficacies in progressive and relapsed AML, but the duration of the clinical response is short. Biomarkers are of great importance in predicting the biological behavior of AML, as well as monitoring the efficacy of FLT3 inhibitors in patients. The expression of biomarkers can be used to predict the disease activity in real time. They can also serve as a guide for administering a particular FLT3 inhibitor to achieve CR and high OS.

Additionally, to overcome the resistance to FLT3 inhibitors caused by different mechanisms, combination therapies of FLT3 inhibitors with other targeted agents or immunotherapeutic approaches are additional areas of investigation.

## Figures and Tables

**Figure 1 cancers-14-01164-f001:**
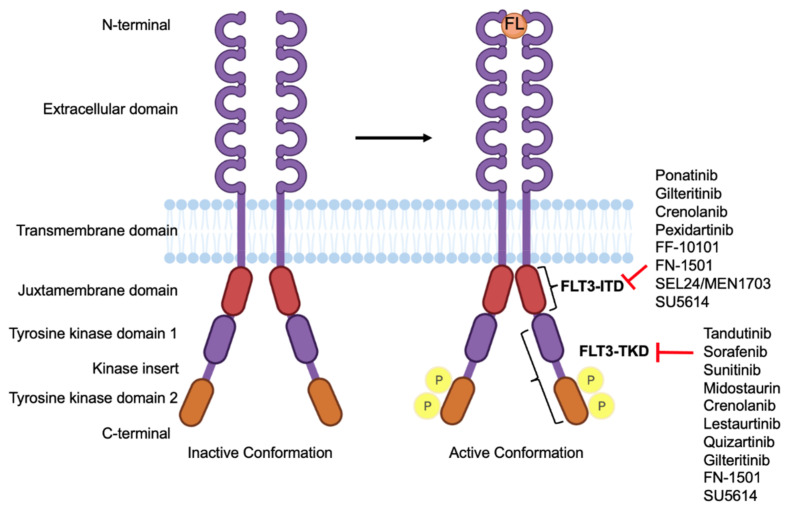
Structure of FLT3 and its drug targets. The structure of FLT3 in its inactive conformation which, upon binding to FLT3 ligand (FL), becomes active, resulting in its autophosphorylation. Different FLT3 inhibitors and their binding sites on their domains.

**Figure 2 cancers-14-01164-f002:**
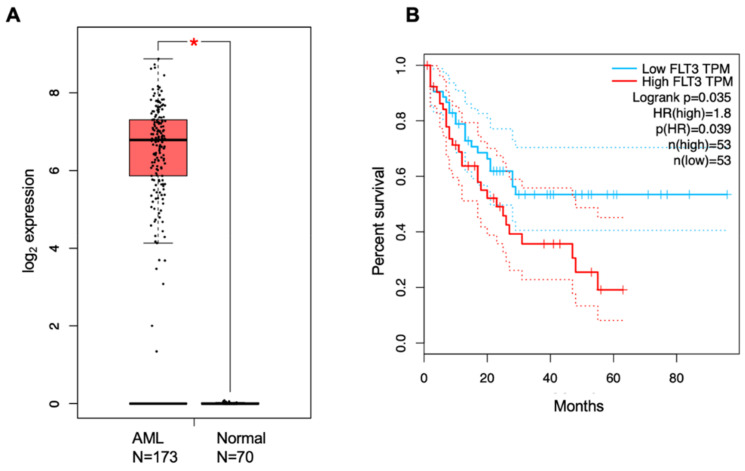
Analysis of FLT3 expression in AML patients. (**A**) Transcript levels of FLT3 in AML patients versus control patients. (**B**) Percent survival of high-FLT3-expressing patients versus low-FLT3-expressing AML patients. The hazard ratio is 1.8, and the *p*-value is 0.035, as analyzed from the TCGA dataset upon GEPIA analysis.

**Figure 3 cancers-14-01164-f003:**
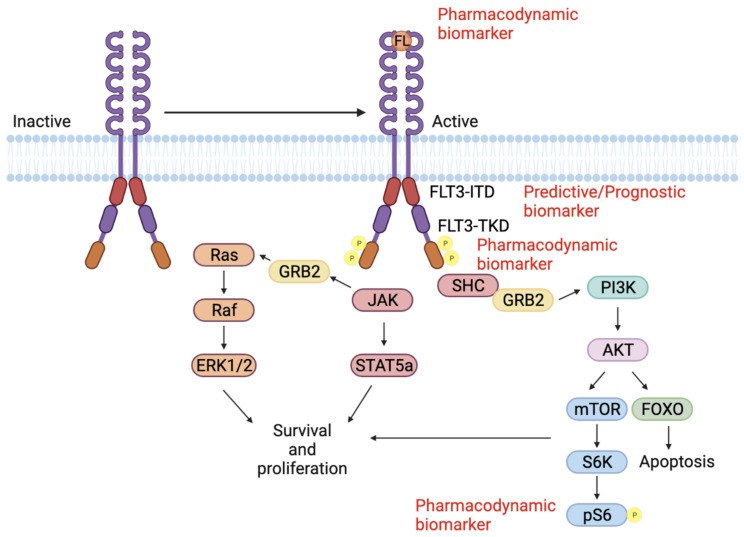
FLT3 signaling pathway. FL binds to the FLT3 receptor and induces receptor dimerization and conformational changes. FLT3 autophosphorylation activates intracellular signaling cascades including RAS/RAF/MAPK PI3K/AKT/mTOR and JAK/STAT. These pathways control cell proliferation, survival, and apoptosis. These different proteins can be used as predictive, prognostic, and pharmacodynamic biomarkers.

**Table 1 cancers-14-01164-t001:** Acute myeloid leukemia and acute leukemias of ambiguous lineage (WHO, 2017).

WHO Classification of Acute Myeloid Leukemia with Recurrent Genetic Abnormalities
AML with recurrent genetic abnormalities
AML with t(8;21)(q22;q22.1); RUNX1-RUNX1T1
AML with inv(16)(p13.1q22) or t(16;16)(p13.1;q22); CBFB-MYH11
APL with PML-RARA
AML with t(9;11)(p21.3;q23.3); MLLT3-KMT2A
AML with t(6;9)(p23;q34.1); DEK-NUP214
AML with inv(3)(q21.3q26.2) or t(3;3)(q21.3;q26.2); GATA2, MECOM
AML (megakaryoblastic) with t(1;22)(p13.3;q13.3); RBM15-MKL1
Provisional entity: AML with BCR-ABL1
AML with mutated NPM1
AML with biallelic mutations of CEBPA
Provisional entity: AML with mutated RUNX1

**Table 2 cancers-14-01164-t002:** FLT3 inhibitors and their targets.

FLT3 Inhibitor	Generation	Type	Target FLT3 Mutations	Other Targets	Phases of Development	References
Sunitinib	First	I	ITD, TKD	VEGFR1, VEGFR2, PDGFRα/β, KIT, RET, CSF1R	II	[85]
Lestaurtinib	First	I	ITD, TKD	JAK2/3, TrkA,B,C AURKA, AURKB,	III	[86]
Midostaurin	First	I	ITD, TKD	EGFR2, KIT, PDGFR, PKCα, VEGFR, Akt	Approved	[87]
Crenolanib	Second	I	ITD, TKD	PDGFRα/β	II	[88]
Gilteritinib	Second	I	ITD, TKD	ALK, AXL	Approved	[89]
Sorafenib	First	II	ITD	VEGFR, PDGFR, c-Kit and RET, RAF	III	[90]
Quizartinib	Second	II	ITD	PDGFRα/β, RET, Kit, CSF1R	II	[91]
Ponatinib	First	II	ITD	Abl, c-Kit, c-Src, FGFR1, PDGFRα, VEGFR2, LYN	Ib/II	[92]
Pexidartinib	First	II	ITD	KIT, CSF1R	I/II	[93]
Tandutinib	First	II	ITD	PDGFRα/β, c-KIT	I	[94]
FF-10101	Novel	I	ITD, TKD	N/A	I	[95]
FN-1501	First	I	ITD, TKD	CDK4/6, KIT, PDGFR, ALK, RET	I	[81]

**Table 3 cancers-14-01164-t003:** Biomarkers for FLT3 AML.

Types	Categories	Markers
Diagnostic biomarkers	Morphological	Bone marrow smears from blast cells
	Immunophenotyping	Early hematopoiesis-associated antigens (CD34, CD38, CD117, HLA-DR)
	Gene fusion	RUNX1-RUNX1T1, CBFB-MYH11, MLLT3-MLL, DEK-NUP214
	Micro RNAs	miR-10a-5p, miR-93-5p, miR-129-5p, miR-155-5p, miR-181b-5p, miR-320d, let-7b, miR-128a, miR-128b, miR-223, miR-142-3p, miR-29a, miR-424, miR-155
Predictive biomarkers	Gene mutations	CEBPA, DDX41, RUNX1, GATA2, ETV6, ANKRD26, NPM1, FLT3
	Long non-coding RNAs	lncRNA RP11-342 M1.7, lncRNA CES1P1, lncRNA AC008753.6
Prognostic biomarkers	Gene mutations	FLT3, NPM1, CEBPA, IDH1/2, DNMT3A, KIT, TP53, RUNX1, and ASLXL1
	FLT3 mutations	FLT3-ITD, FLT3-TKD
	Protein	FLT3 ligand
Pharmacodynamic biomarkers	Protein	Phospho-FLT3 Phospho-S6 Follistatin
	Immune markers	CD112, CD155

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
