# Peer review of "Role of Biomarkers in FLT3 AML"

_cancers, 2022, doi:10.3390/cancers14051164_

Round 1

Reviewer 1 Report

 In this article, the authors described in detail the role of biomarkers in FLT3 AML. Generally, this article is reasonably written and may be informative and instructive for hematologists and oncologists. However, the manuscript contains a number of points to be corrected.

Minor comments:

  1. The subtitle No.7, Pharmacodynamic biomarkers may be inadequate for the paragraph and should be changed such as “Indicators for the efficacy of FLT3 inhibitors”.
  2. The full terms followed by abbreviation is rather confusing throughout the manuscript; duplication of the definition/abbreviation, no definition, and abbreviation without the parenthesis (line 263).
  3. Please appropriately explain about FLAT3-TKD, especially FLT3-ITD, in a paragraph in page 2, line 45-60.
  4. Figure 2 in page 2, line 52 → Figure 1
  5. Figure 3 in page 2, line 64 → Figure 2
  6. The sentence (line 82-84) should be corrected.
  7. The sentence (line 160-162) should be corrected.
  8. Midstaurin, Gilteritinib, Quizartinib, etc, for example, in a sentence in page 6, line 201 should be midstaurin, gilteritinib, quizartinib in all cases.
  9. Inter-mediate in page 8, line 274→intermediate
  10. AR ratio in page 8, line 275 → AR
  11. 100000 in page 1, line 33→100,000

   x

Author Response

In this article, the authors described in detail the role of biomarkers in FLT3 AML. Generally, this article is reasonably written and maybe informative and instructive for hematologists and oncologists. However, the manuscript contains a number of points to be corrected.

We thank the reviewer for their comprehensive assessment of our manuscript. Below are our point-by-point responses:

Minor comments:

1. The subtitle No.7, Pharmacodynamic biomarkers may be inadequate for the paragraph and should be changed such as “Indicators for the efficacy of FLT3 inhibitors”.

We thank the reviewer for their comment. We have changed the title as per the reviewer’s comment.

2. The full terms followed by abbreviation is rather confusing throughout the manuscript; duplication of the definition/abbreviation, no definition, and abbreviation without the parenthesis (line 263).

 We thank the reviewer for their comment. We have fixed this error throughout the manuscript.

3. Please appropriately explain about FLAT3-TKD, especially FLT3-ITD, in a paragraph in page 2, line 45-60.

We thank the reviewer for pointing this out. We have explained FLT3-ITD in lines 69-75 and FLT3-TKD in lines 75-79.

4. Figure 2 in page 2, line 52 → Figure 1

 We thank the reviewer for pointing this out. We have corrected this error.

5. Figure 3 in page 2, line 64 → Figure 2

We thank the reviewer for pointing this out. We have corrected this error.

6. The sentence (line 82-84) should be corrected.

We thank the reviewer for pointing this out. We have corrected the sentence.

7. The sentence (line 160-162) should be corrected.

We thank the reviewer for pointing this out. We have corrected the sentence.

8. Midostaurin, Gilteritinib, Quizartinib, etc, for example, in a sentence in page 6, line 201 should be midostaurin, gilteritinib, quizartinib in all cases.

We thank the reviewer for pointing this out. We have corrected this error throughout the manuscript.

9. Inter-mediate in page 8, line 274→intermediate

We thank the reviewer for pointing this out. We have corrected this typo.

10. AR ratio in page 8, line 275 → AR

We thank the reviewer for pointing this out. We have corrected this error.

11. 100000 in page 1, line 33→100,000

 We thank the reviewer for pointing this out. We have corrected this error.

Reviewer 2 Report

Nitika et al submitted a review entitled « Role of biomarkers in FLT3 AML ». This review is well written and provides an original overview of the literature in the field.

Page 9 line 339 to page 10 line 351: littérature references are lacking for this part of the review

Page 2 line 52: Figure 1 instead Figure 2

Page 2 line 64 : Figure 2 instead Figure 3

Table 3 is not mentioned in the text

Author Response

Nitika et al submitted a review entitled « Role of biomarkers in FLT3 AML ». This review is well written and provides an original overview of the literature in the field.

We thank the reviewer for their comprehensive assessment of our manuscript. Below are our point-by-point responses:

Page 9 line 339 to page 10 line 351: littérature references are lacking for this part of the review

We thank the reviewer for pointing this out. We have now added the appropriate references.

Page 2 line 52: Figure 1 instead Figure 2

We thank the reviewer for pointing this out. We have corrected this error.

Page 2 line 64 : Figure 2 instead Figure 3

We thank the reviewer for pointing this out. We have corrected this error.

Table 3 is not mentioned in the text

We thank the reviewer for pointing this out. We have mentioned table 3 in the text now.

Reviewer 3 Report

I congratulate Nitika and co-workers for this manuscript (cancers-1593464). It is well written and interesting in its first version. However, I would suggest some modifications in order to improve it.

Major revisions:

Section 3 is too short. It would be interesting to further explain the biomarkers mentioned in this section as well as their use in tracking FLT3 inhibition in AML.

The order of the sections doesn’t make sense, I would put biomarkers sections 3 to 6 before actual section 2. First diagnostic, then treatment.

Minor revisions:

There are several small changes to the text that are suggested in the commented (attached) manuscript. These are minor corrections to English, but as mentioned before, the text is well written.

I would suggest homogenizing the nomenclature of Inhibitors and Compounds mentioned in this review. Sometimes the names are started with capitals, but not always. I would change the start of all those names to lower case letters because this is the common usage in scientific publications (unless they are placed after a full-stop, of course).

Author Response

I congratulate Nitika and co-workers for this manuscript (cancers-1593464). It is well written and interesting in its first version. However, I would suggest some modifications in order to improve it.

We thank the reviewer for their comprehensive assessment of our manuscript. Below are our point-by-point responses:

Major revisions:

Section 3 is too short. It would be interesting to further explain the biomarkers mentioned in this section as well as their use in tracking FLT3 inhibition in AML.

 We thank the reviewer for their comment. We agree with the reviewer and have now made the changes suggested by the reviewer in the revised manuscript. 

The order of the sections doesn’t make sense, I would put biomarkers sections 3 to 6 before actual section 2. First diagnostic, then treatment.

 We thank the reviewer for their comment. We agree with the reviewer and have now made the changes suggested by the reviewer in the revised manuscript. 

Minor revisions:

There are several small changes to the text that are suggested in the commented (attached) manuscript. These are minor corrections to English, but as mentioned before, the text is well written.

We thank the reviewer for their comments. We have incorporated all the changes suggested by the reviewer.

1. This section is too short. It would be interesting to further explain the biomarkers mentioned and their use in tracking FLT3 inhibition in AML.

We thank the reviewer for this suggestion. We agree with the reviewer and have now added more information to this section on Biomarkers for FLT3 inhibition in AML.

2. I would put all biomarkers sections (3 to 6) before actual section 2. First diagnostic, then treatment.

 We thank the reviewer for this suggestion. We have now placed sections 3 to 6 before section 2 in the revised manuscript. Figure 2 has been moved in section 7 and changed to Figure 3.

3. Is sorafenib a type II inhibitor? It is a wide-spectrum kinase inhibitor.

Yes, sorafenib is classified as a type II inhibitor. Type II FLT3-inhibitors are only able to bind the inactive FLT3 receptor.

 4. 3 to 400 or 300 to 400, please clarify.

 3 to 400 base pairs as stated in Tsai et al., 2020 (reference-66)

I would suggest homogenizing the nomenclature of Inhibitors and Compounds mentioned in this review. Sometimes the names are started with capitals, but not always. I would change the start of all those names to lower case letters because this is the common usage in scientific publications (unless they are placed after a full-stop, of course).

We thank the reviewer for pointing this out. We have corrected this error throughout the manuscript.